# Effect of Hydrogen Nanobubbles on the Mechanical Strength and Watertightness of Cement Mixtures

**DOI:** 10.3390/ma14081823

**Published:** 2021-04-07

**Authors:** Won-Kyung Kim, Young-Ho Kim, Gigwon Hong, Jong-Min Kim, Jung-Geun Han, Jong-Young Lee

**Affiliations:** 1School of Civil and Environmental Engineering, Urban Design and Study, Chung-Ang University, Seoul 06974, Korea; kwonk2004@naver.com (W.-K.K.); younghogeo@naver.com (Y.-H.K.); 2Department of Civil and Disaster Prevention Engineering, Halla University, 28 Halladae-gil, Wonju-si, Gangwon-do 26404, Korea; g.hong@halla.ac.kr; 3School of Mechanical Engineering, Chung-Ang University, Seoul 06974, Korea; 0326kjm@cau.ac.kr; 4Department of Intelligent Energy and Industry, Chung-Ang University, Seoul 06974, Korea

**Keywords:** hydrogen nanobubble water (HNBW), cement mixture, mechanical strength, watertightness

## Abstract

This study analyzed the effects of applying highly concentrated hydrogen nanobubble water (HNBW) on the workability, durability, watertightness, and microstructure of cement mixtures. The number of hydrogen nanobubbles was concentrated twofold to a more stable state using osmosis. The compressive strength of the cement mortar for each curing day was improved by about 3.7–15.79%, compared to the specimen that used general water, when two concentrations of HNBW were used as the mixing water. The results of mercury intrusion porosimetry and a scanning electron microscope analysis of the cement paste showed that the pore volume of the specimen decreased by about 4.38–10.26%, thereby improving the watertightness when high-concentration HNBW was used. The improvement in strength and watertightness is a result of the reduction of the microbubbles’ particle size, and the increase in the zeta potential and surface tension, which activated the hydration reaction of the cement and accelerated the pozzolanic reaction.

## 1. Introduction

The construction industry consumes large quantities of fossil fuels and accelerates environmental pollution and global warming by continuously emitting carbon dioxide (CO_2_) [1]. About 700–900 kg of CO_2_ are generated when one ton of cement is produced, and the demand and production of cement are expected to increase continuously, owing to continuous infrastructure development [2,3,4]. CO_2_ has been recognized as a major cause of global warming, as has the abnormal climate caused by the greenhouse effect because it accounts for the largest proportion of greenhouse gases [5,6]. It has been reported that concrete corrosion increases by about 15% when global temperature increases by 2 °C, owing to the abnormal climate [7]. The increasing CO_2_ content in the atmosphere due to the temperature increase accelerates carbonation, which reduces the durability of concrete structures. Moreover, the reduction of cement content in concrete structures reduces its production in cement plants. In turn, the lower volume of cement production significantly reduces CO_2_ emissions during the technological process, which is a beneficial pro-ecological action. In addition, eco-friendly/high-performance materials need to be developed in preparation for the decrease in the durability of concrete structures owing to global warming. 

To this end, studies on alternative (substitute) materials and durability-enhancement methods for reducing cement usage have recently been conducted. Granulated blast-furnace slag, ultrafine granulated copper slag, and porous feldspar have been used as cement-replacement materials to reduce cement content while improving its mechanical properties [8,9,10]. 

The strength of a cement mixture can be increased through reinforcement mixing, which includes adding materials such as steel fiber, glass fiber, polymeric fiber, carbon fiber, and carbon nanotubes (CNTs). Recently, many studies have been conducted to improve cement’s mechanical properties (compression, flexure, and durability) by adding a small amount of graphene oxide (GO), which is a nano-fusion material [11]. GO inhibits the diffusion of cracks owing to its high tensile properties, and activates the hydration reaction by providing a reaction surface where cement hydrates (C-S-H and Ca(OH)_2_) can react to form strong covalent bonds [12,13]. 

Another method for activating the cement-hydration reaction is by adding microbubbles to the mixing water. When a large number of microbubbles is added, the compressive strength of cement is improved compared to that when using general water [14,15]. Recently, the cement content was reduced while the compressive strength was improved by applying hydrogen nanobubble water (HNBW) as the mixing water for a cement mixture. It was determined that the strength increased because the zeta potential and vibration characteristics of the surface of the hydrogen nanobubble (HNB) accelerated the hydration reaction, thereby increasing the density of the cement matrix [16].

Nanobubbles are defined as microbubbles with a diameter of less than 200 nm [17]. A nanobubble is almost unaffected by buoyancy, lasts for a long time in water owing to its nanoscale size, has a wide specific surface area and high internal pressure, and generates radicals within the interface [18,19]. These special properties are useful in various fields, such as surface coating and cleaning, contaminant removal, improving the flotation of ultrafine minerals, and releasing energy when the bubbles are destroyed [20,21,22,23]. 

In this study, the hydration reaction was activated using microbubbles in the mixing water in order to improve the mechanical properties of cement mortar. Moreover, the effect of the number of nano-sized microbubbles on the activation of the hydration reaction was analyzed. Osmosis was used to prepare HNBW with a large amount of HNBs. Changes in the workability, strength, and internal microstructure were observed by applying HNBW with a high HNB concentration as the mixing water of a cement mixture. The purpose of this study is to investigate the effect of mixing water with a high HNB concentration on the development of cement hydration reactants, pore structure, and watertightness of the hardened cement mixtures. 

## 2. Experimental Setup 

### 2.1. Hydrogen Nanobubble Water (HNBW)

#### 2.1.1. HNBW Production Method 

A capillary with a decompression method was applied for HNBW, as shown in Figure 1. The bubbles were generated using hydrogen gas, after filling the water tank with general water filtered with a 5 μm filter (Purity: 99.995%, Shinyoung Gas Co., Seoul, Korea). The hydrogen gas was injected at 0.5 MPa through a ceramic filter; it was then injected into the top of the water tank by maintaining the internal tank pressure at 0.15 MPa. The pressurization was continued for 24 h to increase the solubility of the gas and improve the bubbles’ stability. The size, distribution by diameter, and bubble concentration in the prepared HNBW were measured using nanoparticle tracking analysis (NTA, LM10, Malvern Panalytical, Westborough, MA, USA).

#### 2.1.2. High-Concentration Nanobubbles with Osmosis

A schematic diagram of the change in diameter according to the osmosis course is shown in Figure 2. Physical equilibrium is achieved when the external pressure acting on the gas/liquid interface of the bubble and the pressure inside the bubble are equal. Micro-sized bubbles are in a metastable state owing to the low internal pressure (*P_in_*) when a high osmotic pressure (P_out_) acts on the bubble surface. Some of the microbubbles maintain a thermodynamic equilibrium by shrinking or disappearing because of the P_out_, which is increased by osmotic pressure [24]. On the other hand, nano-sized bubbles remain because high osmotic pressure is favorable for their survival. It is believed that HNBs will be stably present, even in a densely concentrated state, because the *P_in_* of the bubbles increases as their size decreases and their concentration increases, thereby increasing the solubility of the gas in water [25,26].

In this study, the HNBs were concentrated using a semipermeable membrane between the two solutions with different concentrations [27]. An osmotic phenomenon was used, in which solvent molecules selectively flow across a semipermeable membrane from a diluted solution to a more concentrated solution. In addition, 60 wt% of PEG (polyethylene glycol, MW = 20,000, CP, Daejung, Korea) was used for the hypertonic solution and the HNBW, as described in Section 2.1, was used for the hypotonic solution. The osmosis was sustained for 40 min (HNBW 40) and 80 min (HNBW 80) by filling the semipermeable membrane with HNBW, sealing the upper and lower parts, and placing it in a PEG solution.

#### 2.1.3. HNBW Analysis Method 

The long-term stability, particle-size distribution, zeta potential of the bubble surface, and surface tension were measured in an experiment for the physicochemical analysis of the prepared HNBs. Nanoparticle tracking analysis and a zeta potential analyzer (ZetaPALS, Brookhaven Corp., Holtsville, NY, USA) were used to determine the long-term HNB stability and particle-size distribution. A Pt plate-type analyzer (DST-60, Surface Electro Optics Co., Suwon, Korea) was used to determine the surface tension.

### 2.2. Experimental Conditions and Method

#### 2.2.1. Cement Mixture Proportions

The specimens were prepared and classified into cement mortar and paste, as shown in Table 1, according to the purpose of the experiment. The cement mortar specimen was used to observe the change in compressive strength and workability. The cement paste specimen was not mixed with sand in order to analyze the reaction characteristics of HNBW and cement in detail. These specimens were used for porosity and microstructure analyses and were prepared according to the ASTM C109/C109M-20b standard [28].

#### 2.2.2. Mortar Flowability and Strength 

The flowability test was conducted immediately after mixing, according to ASTM C1437, to analyze the effect of the HNB concentration on the workability of the cement mortar by measuring the average diameter of the mortar [29]. In addition, the mechanical properties were evaluated by measuring the compressive strength of the cement mortar, according to the HNB concentration. Three specimens were prepared for each condition in a 50 × 50 × 50 mm^3^ mold, according to ASTM C109, demolded after 24 h, and cured for 3, 7, 14, and 28 days in water at 24 °C. For the compressive strength measurement, they were compressed and fractured at a rate of 1 mm/min using a universal testing machine (UTM, HJ-1295, Heungjin Testing Machine, Gimpo, Korea) and analyzed by averaging the three compressive strengths for each condition.

#### 2.2.3. Paste Porosity and Microstructure 

We used mercury intrusion porosimetry (MIP) to analyze the effect of the HNBW on the pore structure (pore size, porosity) of the cement mixtures. A MicroActive AutoPore V 9600 porosimeter (Micromeritics Instrument Corp., Norcross, GA, USA) was used as an analyzer. The specimens were prepared by curing in water for seven days under the conditions in Table 1. Then, 0.5–1.0 g of the specimens was used for analysis, after being immersed in acetone for five days to prevent weathering. A scanning electron microscope (SEM, S-3400N, Hitachi, Tokyo, Japan) image was taken to analyze the change in microstructure according to the HNB concentration, and the same specimen was used as for the MIP analysis.

## 3. Experimental Results

### 3.1. HNBW

Figure 3a shows the result of measuring the HNBW bubble concentration, prepared as described in Section 2.1.1, using NTA. It shows that the bubble concentration was 0.89 × 10^8^ particles/mL after 35 days, with repeating variations, thereby indicating that about 80% of the initial value survived. The results of measuring the bubble diameter over time are as shown in Figure 3b. It remained stable for 35 days, while repeating the decrease and variation, thereby demonstrating the suitability of the HNBW used in the experiment. 

The HNB particle size and concentration according to osmosis time are shown in Figure 4, which shows that the maximum particle diameter of HNBW(0) without osmotic pressure was distributed up to about 600 nm. The maximum particle diameters of HNBW(40) and HNBW(80), where the osmotic pressure was applied for 40 min and 80 min, respectively, were distributed to about 440–480 nm. The HNB distribution decreased to below 200 nm as the osmosis duration increased. The results of the NTA measurement of the diameter and the total HNB concentration according to osmosis time are summarized in Table 2. 

The surface charge of the bubbles was concentrated, and a negative zeta potential was found when the gas was compressed. The internal pressure (P_i_) increased as the bubble diameter decreased and equalized. The zeta potential was used as a criterion for inter-particle stability. This is because the strong electrostatic repulsion between bubbles caused by the zeta potential prevents the agglomeration of bubbles and disperses them stably in water [30]. Moreover, it resists high internal pressure when the surface tension increases as the bubble diameter decreases [31]. The zeta potential and surface tension measurement results by HNB concentration are as shown in Figure 5, which shows that the (−) zeta potential and surface tension increased with the bubble concentration. The (−) zeta potential increased because a high charge density was formed on the surface of the bubble as the gas was compressed, and P_i_ rose when the bubble diameter decreased. In other words, it is believed that the HNBs maintained high stability because a high surface tension was formed, owing to the strong hydrogen bonds at the gas–liquid interface of the HNBs.

### 3.2. Cement Mortar 

#### 3.2.1. Flowability 

When microbubbles are used in mixing water, the workability is reduced and the durability is improved because a larger amount of free water reacts with the cement, owing to its large specific surface area [14,15,32,33]. The results of the flowability test showed that the diameter of EXP-GW, which used general water, was about 159 mm, and the bubble diameter for EXP-H(40) and EXP-H(80), with concentrated HNBs, was about 150 mm and 143 mm, respectively, showing that the diameter gradually decreased as the HNB concentration increased. The formation of a homogeneous cement mixture can be expected because the decrease in the mortar workability as the HNBs are concentrated increases the reaction between the cement and water.

#### 3.2.2. Mechanical Properties

The results of the compressive strength measurement for each curing time in Table 1 are shown in Figure 6a. The compressive strength from 3 to 28 days of curing was 23.66–35.68 MPa when general water was used (EXP-GW), and it was 25.98–37.0 MPa and 28.38–38.67 MPa for the specimens that used HNBW(40) and HNBW(80) as mixing water, respectively. This shows that the compressive strength improved as the HNB concentration increased. 

Figure 6b shows that the rate of strength increase was higher when concentrated HNBW was used, compared to when general water was used (EXP-GW). EXP-H(40) improved by about 4.58–7.99%, and the specimen that used HNBW(80) improved by about 8.36–15.79%. In addition, the 7-day-cured specimens of EXP-H(40) and EXP-H(80) developed 88–90% of their total 28-day strength, thereby indicating that HNBs are associated with early strength development.

#### 3.2.3. Pore Structure Characteristics

In this study, the pores of the cement paste were classified into gel pores (<10 nm), capillary pores (10 nm–1 μm), large capillary pores (1–100 μm), and entrained air voids (>100 μm) [34,35]. The porosity, average pore size, and density from the MIP test are shown in Table 3. It shows that the porosities of the cement paste were 41.33%, 36.94%, and 31.06%, for EXP-GW, EXP-H(40), and EXP-H(80), respectively, which gradually decreased as the HNB concentration of the mixing water increased. The diameters of the pores decreased by about 10 nm and the density increased as the porosity decreased. 

The cumulative pore volume according to the pore diameter is shown in Figure 7a. It shows that the cumulative pore volume of EXP-GW was about four times larger than that of EXP-H(80). This means that a denser pore structure was formed as a smaller number of bubbles was introduced during the cement paste mixture when the HNB concentration was higher. 

Figure 7b shows an increase in pore volume of below 1 μm in a differential value. The curves of all specimens are in the form of a bimodal distribution with two peaks, in which the first peak is in the range of 10–20 nm and the second peak is in the range of 100–200 nm. The interval between the two peaks differs in each case. The interval between the peaks is small in EXP-GW, whereas the peak value is relatively decreased as the capillary pores (10 nm–1 μm) are widely distributed in EXP-H(40). However, the interval decreases as the peak value increases for EXP-H(80). Figure 7c shows an accurate distribution, according to the size of the pores inside the cement paste, based on the HNB concentration. The total porosity gradually decreased with the change in the internal pore distribution. In particular, it was determined that the rapid decrease in the entrained porosity greatly influenced the porosity. The proportion of relatively large pores of 1 μm or above rapidly decreased to about 40%, 30%, and 15%, respectively, while the gel porosity smaller than 10 nm increased slightly to about 6.8%, 8.8%, and 12.3%. This change in pore distribution means that a tight structure formed in the cement paste.

Figure 8 shows a SEM image of the cement paste in each case. A thin and low-density ettringite, C-S-H crystals in the early formation stages, and unhydrated cement particles were observed in EXP-GW. Overall, the internal pores were widely developed because the hydration reactant developed more slowly than that of EXP-H(40). It was found that the internal structure of the cement paste became denser as the ettringite thickened with the increase in the HNBW concentration, and developed along with the C-S-H crystals in EXP-H(40) and EXP-H(80). In addition, portlandite (CH) was actively formed, owing to the development of a hydration reaction, according to the increase in HNB concentration, while the crystal phase gradually changed to an irregular shape owing to the activation of the pozzolanic reaction. This is believed to be because a large amount of CH was consumed to form the C-S-H gel, owing to the acceleration of the pozzolanic reaction. 

This difference in the degree of hydrate development was also found through the bimodal curve (Figure 7b), as a result of the MIP test. In the case of EXP-H(40), it was determined that the pore-size distribution was expanded to 1 μm because CH with a size of 0.6–2 μm was developed. On the other hand, the interval between the peaks was narrower and the relative strength of the peaks increased in the case of EXP-H(80), because C-S-H with less than 0.1 μm was developed.

## 4. Discussion

The compressive strength increased because the density increased as the pores of the cement mixture decreased with the activation of the hydration reaction. It was determined that the hydration reaction was activated by the characteristics that contribute to the stability of HNBs, as shown in Section 3.1. This is explained in detail by dividing the aspects of the collision impact between HNBs and cement particles, and the influence of the surface tension and zeta potential of the bubble.

### 4.1. Collision Impact of HNBs and Cement Particles

A homogeneous size of about 100–150 nm was maintained, as the diameter of the HNBs decreased when their concentration increased, as shown in Table 2. The water flotation increased as the HNBs became smaller, thereby increasing the collision frequency between particles [14,15]. Although the collision possibility is low because the cement particles are 20–30 μm in diameter, HNBs formed on a hydrophobic surface can increase the possibility of collisions with cement particles, owing to the bubble-bridging mechanism. It provides a surface favorable for adhesion to the normal bubbles generated during the cement-mixing process, as shown in Figure 9. Therefore, a homogeneous cement mixture can be formed by activating the remaining cement particles without causing a hydration reaction. In other words, a high density was formed in the internal structure of the cement mixture because the HNBs created an environment favorable for the development of hydration reactants, which enhanced the strength. 

### 4.2. Effect of Bubble Surface Tension and Zeta Potential

The surface tension and (−) zeta potential of the surface increase as the HNB diameter decreases, thereby maintaining high stability in water. The surface tension pulls water into the capillary formed between the particles, in which the capillary height increases when the surface tension is stronger. The capillary height is further increased because the nano-sized capillary formed inside the cement-hydration reactant has an extremely small diameter. Therefore, HNBW is dispersed to the fine structure inside the cement-hydration reactant (e.g., C-S-H), thereby coming into contact with more cement particles. 

In addition, the smaller HNBs move in a Brownian motion and vibrate the surrounding water molecules. The high zeta potential further activates this effect, thereby dispersing the water more widely within the microstructure of the cement mixture. This phenomenon is due to the reduction of the material bleeding caused by the formation of a homogeneous mixture and a tight internal cement-mixture structure promoting the hydration reaction.

## 5. Conclusions 

The effect of applying high-concentration hydrogen nanobubble water on the workability, durability, watertightness, and microstructure of a cement mixture was analyzed in this study, and the following conclusions were obtained as a result of this research:The number of hydrogen nanobubbles was concentrated by more than twice, and the diameter of the concentrated bubbles was stabilized to a uniform size of about 100–150 nm by applying osmosis. When hydrogen nanobubble water was applied as a mixing water, the compressive strength of the cement mortar improved by about 3.7–15.79% compared to the control group that used general water. Moreover, the compressive strength increased as the concentration of hydrogen nanobubbles increased.The compressive strength increased because the hydrogen nanobubbles induced more contact between the mixing water and the cement particles through stabilization. Further, the mixing water reached the microstructures constituting the cement-hydration reactant because the zeta potential and surface tension of the HNBs increased. This led to the activation of the hydration reaction and the acceleration of the pozzolanic reaction, and the cement mortar’s compressive strength and watertightness were improved.The results of the MIP and SEM analyses indicated that the highly concentrated nanobubble water formed the cement mixture into a tighter structure by reducing the development and porosity of the hydration reactant. The pore volume was reduced by up to 75% when highly concentrated HNBW was used as the mixing water. This is because fewer bubbles were introduced when the cement mixture was mixed as the HNB concentration increased, thereby forming a denser pore structure. As a result, the concentration of HNBs increased, the hydration of the cement was actively induced due to the change in the physicochemical properties of the mixing water. Therefore, the tightness of the internal hydrate structure increased the strength and durability of the cement mixtures.

Research on bubble size control and concentration increase will be applicable as a core technology for various nano-applied research topics, such as water purification, growth promotion, sterilization, and medical treatment. In addition, it can be actively used as an eco-friendly construction material that can prevent the deterioration of concrete structure durability, such as preventing the corrosion of reinforcing steel bars.

## Figures and Tables

**Figure 1 materials-14-01823-f001:**
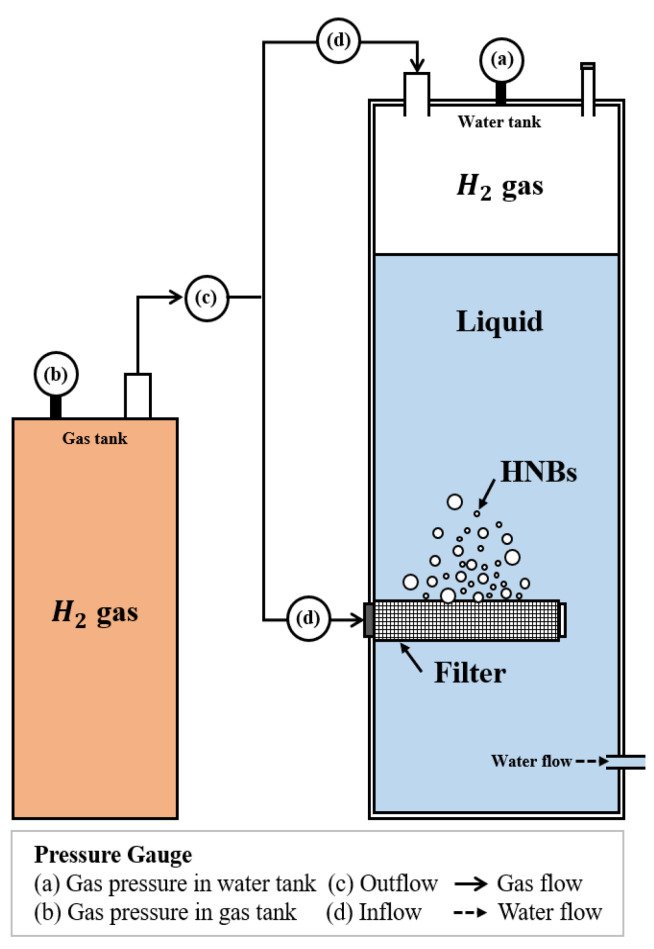
Schematic view of nanobubble generator.

**Figure 2 materials-14-01823-f002:**
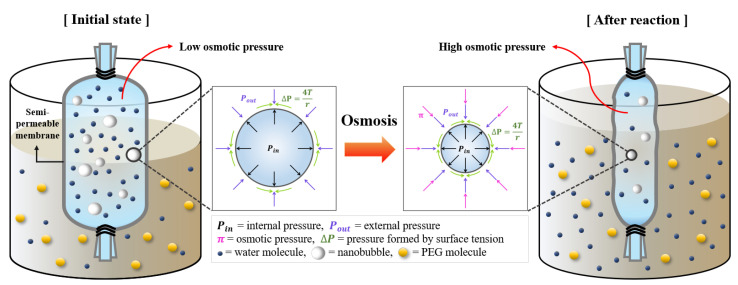
Effects of osmosis on the bubble size and state.

**Figure 3 materials-14-01823-f003:**
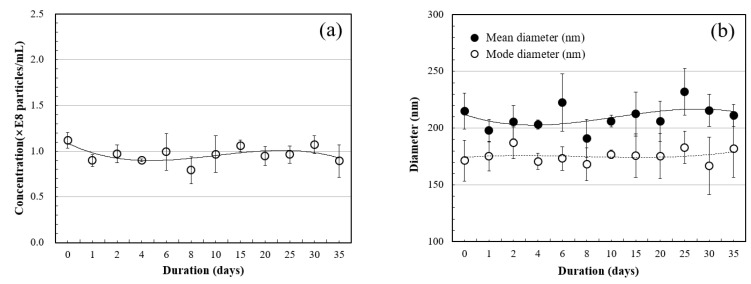
NTA test results: (**a**) average particle concentration of nanobubbles; (**b**) average mean and mode diameter of nanobubbles.

**Figure 4 materials-14-01823-f004:**
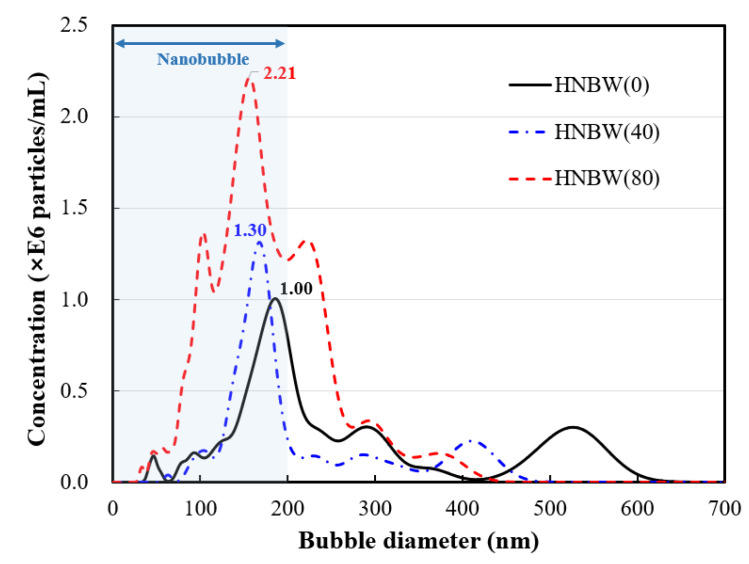
Size distribution of nanobubbles according to high densification. Nanobubbles with a diameter of 200 nm or less are gradually increasing, and this section is marked with a blue background.

**Figure 5 materials-14-01823-f005:**
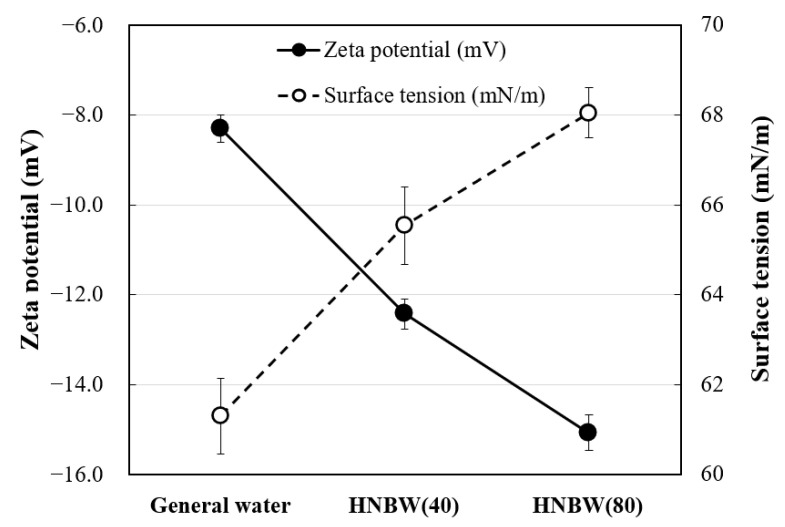
Zeta potential and surface tension according to the concentration of nanobubbles.

**Figure 6 materials-14-01823-f006:**
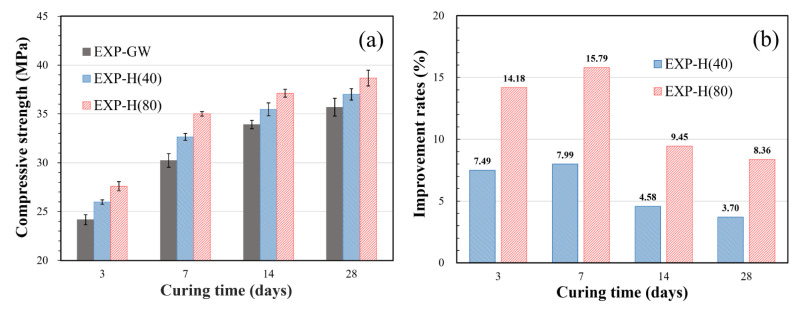
Compressive strength of cement mortar: (**a**) comparison of average compressive strength; (**b**) improvement rates of compressive strength compared to EXP-GW.

**Figure 7 materials-14-01823-f007:**
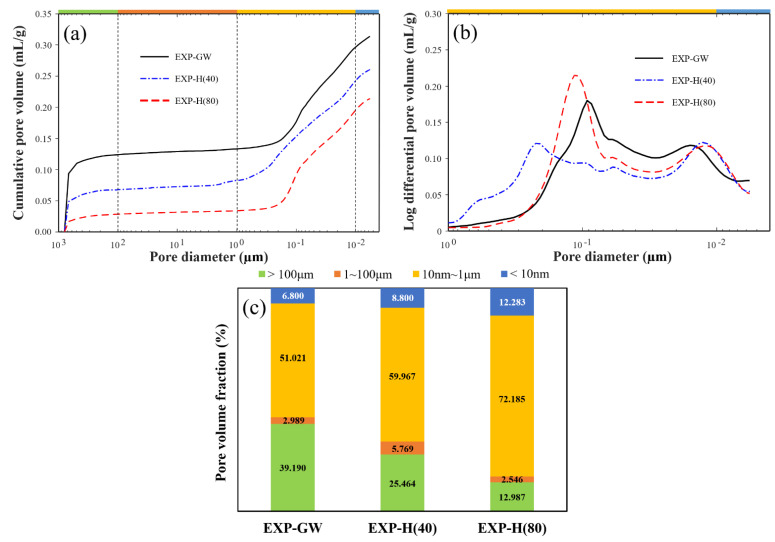
MIP test results: (**a**) Cumulative pore volume according to mercury intrusion; (**b**) log differential pore volume under 1 μm pore diameter; (**c**) pore volume fraction.

**Figure 8 materials-14-01823-f008:**
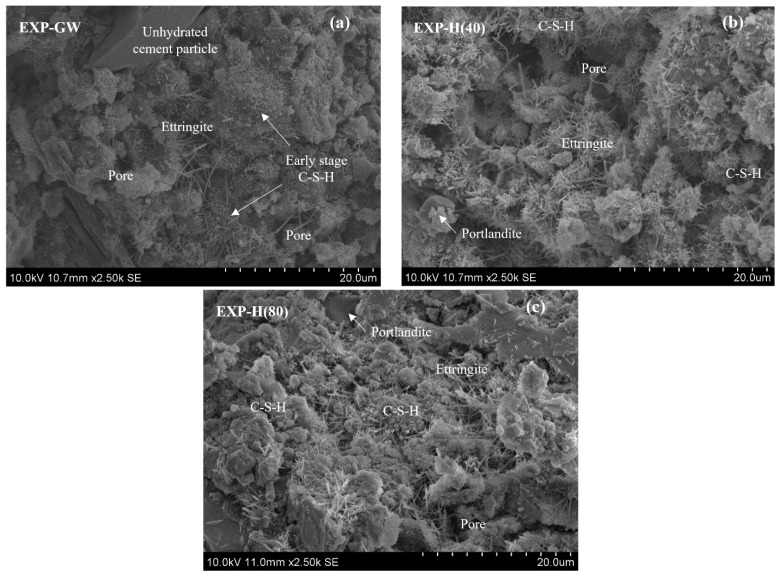
SEM images of cement paste after 7 days of curing: (**a**) EXP-GW; (**b**) EXP-H(40); (**c**) EXP-H(80).

**Figure 9 materials-14-01823-f009:**
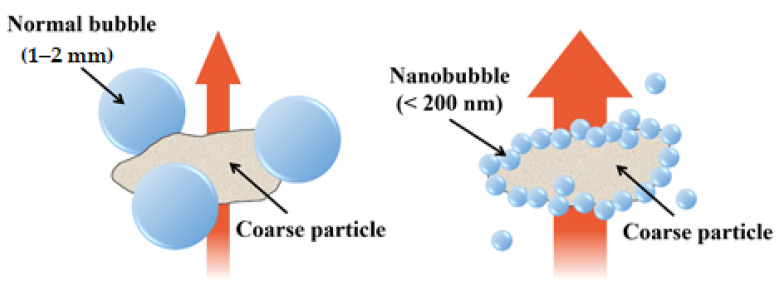
Effect of nanobubble size on a cement particle’s flotation [36].

**Table 1 materials-14-01823-t001:** Experimental conditions (fresh mix).

Mixtures	Mixing Ratio (%)	EXP-GW	EXP-H(40)	EXP-H(80)	Curing Time (days)	Test Method *
Cement mortar	25.4	Ordinary Portland cement (OPC)			3, 7, 14, 28	①, ②
	62.2	Sand				
	12.4	General water	HNBW(40) ^a^	HNBW(80) ^b^		
Cement paste	67.2	Ordinary Portland cement (OPC)			7	③, ④
	32.8	General water	HNBW(40)	HNBW(80)		

* ① Flow test, ② Compression strength test, ③ MIP (mercury intrusion porosimetry), ④ SEM (scanning electron microscopy); ^a^ HNBW sustained osmosis for 40 min, ^b^ HNBW sustained osmosis for 80 min.

**Table 2 materials-14-01823-t002:** NTA test results of HNBW according to osmosis duration.

Properties	HNBW(0)	HNBW(40)	HNBW(80)
Osmosis duration (min)	-	40	80
Total concentration (particles/mL)	131.12 × 10^6^	150.29 × 10^6^	275.94 × 10^6^
Mean diameter (nm)	276	181	182
Mode diameter (nm)	185	158	156

**Table 3 materials-14-01823-t003:** MIP test results (porosity, average pore diameter, and bulk density).

Case	Porosity (%)	Average Pore Diameter (nm)	Bulk Density (g/cm^3^)
EXP-GW	41.32	43.19	1.32
EXP-H(40)	36.94	38.50	1.42
EXP-H(80)	31.06	29.93	1.45

## Data Availability

Data sharing is not applicable to this article.

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
