# Peer review of "Effect of Hydrogen Nanobubbles on the Mechanical Strength and Watertightness of Cement Mixtures"

_materials, 2021, doi:10.3390/ma14081823_

Round 1

Reviewer 1 Report

The authors of the article presented research on the effect of the use of highly concentrated nano-bubble hydrogen water (HNBW) on workability, durability, water tightness and microstructure of cement mixtures. As a result, the compressive strength of the cement mortar and water tightness were increased compared to the reference samples.

Unfortunately, there are some uncertainties and doubts in the article, which are presented below. This requires an explanation by the authors. The article is not eligible for publication in Materials as it stands.

General remarks

The conclusions did not explain the impact of the high-concentration hydrogen nano-bubble on the durability of the cement mixture.

Specific remarks

  1. 36 line The sentence should be worded differently: “Increasing the CO2 content in the atmosphere due to the temperature increase accelerates carbonation, which reduces the durability of concrete structures.
  2. 37 line: The sentence should be worded differently: Moreover, the reduction of cement content in concrete structures reduces its production in cement plants. In turn, the lower volume of cement production significantly reduces CO2 emissions during the technological process, which is a beneficial pro-ecological action.
  3. 131 line: What does OPC mean. The abbreviation should be explained.
  4. 139 line: Why was the 24 hour compressive strength test according to ASTMC 109 not performed?
  5. 178 line: Table 2: There should be HNBW (0) instead of HNBW (O).
  6. 197, 204, 205 lines Fig. 5 Lack of consistency in the markings should be on the abscissa axis EXP-HNBW (0) instead of EAP-GW and then HNBW (40) and HNBW (80).
  7. 222 line The compressive strength after 1 day was not determined which would confirm this observation.
  8. 257 line: Fig.7a, b: it should be: on the ordinate axis (ml / g) not (mL / g) Fig.7c in the capillary pores size description it should be (> 100 μm) instead of (> 100 um) etc.

I recommend a moderate review of the manuscript, including comments, to make it an article suitable for publication in the Materials.

Reviewer 2 Report

Please find attached a PDF file with my comments and suggestions for authors.

Reviewer 3 Report

The manuscript „Effect of Hydrogen Nano-Bubbles on the Mechanical Strength 2 and Watertightness of Cement Mixtures” is very interesting, innovative and concerns the influence of hydrogen nano-bubble water (HNBW) used in the production of cement composite.

The authors discussed the method of producing hydrogen nano-bubble water (HNBW) and clearly presented it in graphical diagrams. During the research, three types of HNBW were used: basic and two stabilized by osmosis. The authors examined the influence of HNBW on selected properties of cement composites (flowability of cement mortars, compression strength of cement mortars, MIP (Mercury Intrusion Porosimetry) of cement pastes and SEM (Scanning Electron Microscope) also of cement pastes). The article is written on 14 pages and includes 37 references.

There are some minor shortcomings in the article:

  • Table 1: please add an annotation "fresh mix" after (1) in the last column; it is obvious, but the maintenance time presented earlier does not apply to this study.
  • Line 130: not every reader knows the details of the ASTM C 109 standard, so please add 2-3 sentences describing the sample preparation procedure.
  • Line 134: same as before, add 2-3 sentences describing the procedure specified in ASTM C 1437.
  • Line 148-149: Incomprehensible and probably not entirely true sentence: "The hydration reaction was stopped after the specimens used in the analysis were cured in water for seven days under the conditions in Table 1." – How was hydration stopped? Taking samples out of the water does not mean that hydration is stopped. Moreover, in Table 1, there are the curing time not the curing conditions.
  • Line 212-213: „…for each experimental condition of Table 1…”, it is more about the age of the samples or the curing time, not the curing conditions – they were the same.
  • Line 213-214, 218-219, 324-325 and line 16: The phrase „The compressive strength was 23.66–35.68 MPa when general water was used (EXP-GW), and 25.98–37.0 MPa and 28.38–38.67 MPa for …” is misleading. Initially, I was wondering why the scattering of results is so large. Finally, after reviewing Figure 6, it turned out that the mentioned ranges of strength values apply (in total) to strengths tested after 3, 7, 14 and 28 days. Please correct these sentences to make them more readable.
  • Table 3: change the unit [g/ml] to the SI unit e.g. [g/cm3].

I also have a question (out of the article): do the authors know the costs of using HNBW in practice, how stable HNBW is over time, and whether ready-mixed concrete plants could use ready-made HNBW or would they need to have an installation for its production. I know it's too early for these kinds of questions, but it's extremely interesting for me.
